# Is the effectiveness of policy-driven mitigation measures on carabid populations driven by landscape and farmland heterogeneity? Applying a modelling approach in the Dutch agroecosystems

**Elżbieta Ziółkowska** [1]*, **Aaldrik Tiktak**[2], **Christopher J. Topping**[3]

**1** Jagiellonian University in Kraków, Institute of Environmental Sciences, Kraków, Poland, **2** Department of Water, Agriculture and Food, PBL Netherlands Environmental Assessment Agency, The Hague, The Netherlands, **3** Department of Ecoscience–Biodiversity and Conservation, Århus University, Aarhus C, Denmark

\* e.ziolkowska@uj.edu.pl

**Data Availability Statement:** Original pseudo-anonimized data from the Land Parcel Identification

## Abstract

The growing challenges of protecting biodiversity in agro-ecosystems and maintaining high agricultural productivity has become an important issue within the European Union, shaping both European and national agro-policies. The presented study is part of a broader evaluation of the interim targets of the 2013 Dutch policy plan on sustainable use of pesticides, carried out in 2019 by the PBL (Planbureau voor de Leefomgeving) Netherlands Environmental Assessment Agency. We aimed to assess the effectiveness of selected mitigation measures suggested in the policy plan on non-target terrestrial arthropods using a common carabid beetle *Bembidion lampros* as a model species. We combined dynamic landscape models with detailed agent-based population modelling to simulate impacts of reduction of toxicity of insecticides, reduction of spray drift to the off-crop area, and increase in area of field margins on the beetle population dynamics in ten agricultural landscapes representing different farming systems. Our simulations showed that a shift towards low-risk products should be the priority if the goal is to increase beetle range. To promote local beetle abundance this needs be coupled with increasing amount of field margins in a landscape. Overall, the observed treatment and landscape effects were highly context-specific and therefore we suggest that care is used when defining and interpreting metrics based on population effects of policy measures. This caveat notwithstanding, the use of simulation to assess complex interactions between landscape, ecology and behaviour of species, and policy measures can be a powerful tool supporting innovative policy management. This should include the development of landscape-context specific targets and/or mitigation measures.

System (LPIS) for the Netherlands cannot be shared publicly due to the data sharing agreement restrictions. Digital object-oriented topographic database of the Netherlands (TOP10NL) is available from the Publieke Dienstverlening Op de Kaart (PDOK) service (www.pdok.nl). Landscape generation scripts, script producing landscape inputs for the ALMaSS and other associated files are available in an open repository on GitLab (https://gitlab.com/ALMaSS/almassauxillary/). The results of simulation runs are now available in supplementary data files. The ALMaSS configuration files for simulation scenarios are provided in Appendix C.

**Funding:** EZ work was supported by the PBL Netherlands Environmental Assessment Agency (https://www.pbl.nl/en) through the project "Developing and application of a methodology to assess impacts of pesticides on key ecosystem services" (contract no. 31134493). The funders had no role in study design, data collection and analysis, decision to publish, or preparation of the manuscript.

**Competing interests:** The authors have declared that no competing interests exist.

## 1. Introduction

There is an increased focus within European Union (EU) policies and strategies on protection of biodiversity while maintaining agricultural sustainability. The Green Deal sets targets for reduction in pesticide use by 50% by 2030. Formulated in the Farm to Fork and Biodiversity strategies, this reduction, together with other goals such as having 10% of the utilized agricultural area (UAA) reserved for wildlife, have their roots in the alarming pattern of decrease in abundance, biomass and range of insects in Europe [1–5] and worldwide [6,7]. Trends, although differing with taxa and regions [8], indicate that the most severe declines occur in landscapes dominated by agriculture [9–13].

Insect declines in agricultural landscapes are mainly driven by land-use changes associated with the transition from natural to agricultural lands and intensification of agricultural practices. These include land consolidation and increased pollution by pesticides and fertilizers [6–8,14]. At the same time, maintaining high species diversity and population abundance within agricultural landscapes is crucial for many ecosystem services. Many of these functions are related directly to crop production, such as natural (biological) pest control and pollination [15,16]. Consequently protecting biodiversity in agro-ecosystems has become an important issue at the EU level. For example the EU Biodiversity Strategy for 2030 focuses on halting biodiversity loss and building healthy and environmentally friendly food systems in Europe. This and the Common Agricultural Policy (CAP) provide a general framework, but more and more it is mainly up to member states to implement this framework in their specific contexts. Hence, protection of biodiversity in agricultural landscapes is reflected in national agro-policies, as well as agri-environmental programs. One such example, although pre-dating the Green Deal, is the policy document 'Healthy Growth, Sustainable Harvest' [17], drawn up by the Dutch government. This document sets up a more sustainable crop protection policy in the Netherlands for the 2013–2023 period. The aim is to achieve a sustainable use of pesticides by reducing the risks and impacts of pesticide use on human health and the environment through promoting the use of Integrated Pest Management (IPM). At the same time, the ambition of this policy plan is to 'strengthen the economic prospects for Dutch agriculture and horticulture'.

In 2019, the policy interim targets related to integrated pest management, ecological water quality, drinking water quality, biodiversity, occupational safety, and food safety were evaluated by the PBL Netherlands Environmental Assessment Agency [18]. In terms of biodiversity, the interim target for 2018 was to reduce risks posed by chemical plant protection products to non-target arthropods by reducing the use of hazardous substances in exchange for low-risk products and non-chemical measures, and creation of habitats (refuges) for pollinators and pest control species. The PBL interim evaluation concluded that this target, among others, has not been achieved; in fact the situation has slightly deteriorated between 2013 and 2018 [18]. The high intensity of farming in the Netherlands may be blamed for this situation. The sale of plant-protection products per hectare of UAA is still very high in the Netherlands compared to other EU Member States. According to Eurostat statistics, in 2018 the Netherlands was in third place in terms of both plant-protection products sales (with approximately 5 kg/ha of UAA) and insecticides sales (with approximately 0.9 kg/ha of UAA). In addition, no substantial shift towards 'low-risk' substances has been observed, and the sale of insecticides (which account for the largest share of environmental burden) has not decreased in recent years [19]. The pesticide-related environmental risk is, however, not equally distributed across arable land, but rather largely depends on dominant farm type, being highest in areas where intensively sprayed crop and plant species, such as potatoes, onions, lilies and tulips, are grown [18].

Both spatial distribution of crop and non-crop habitats and management of crop habitats (including applications of pesticides) drive source-sink dynamics and performance of non-

target organisms in complex agricultural landscapes. To secure crucial agro-ecosystem functions, a broader, system-based perspective is needed which considers large-scale effects of plant protection products [20]. Coupling landscape models with agent-based population modelling and detailed knowledge on ecology and ecotoxicology allows simulation and better understanding of population-level impacts of pesticide applications [21,22]. These techniques can be also use for conducting in silico experiments to test management implications and to answer policy questions regarding biodiversity protection in agricultural landscapes [23–26].

The aim of this study was to evaluate the effectiveness of selected mitigation measures from the Dutch policy document 'Healthy Growth, Sustainable Harvest' on non-target terrestrial arthropod, focusing on spring-breeding carnivorous ground beetles (Carabidae) being important for biological control of insect pests. We applied combination of spatial analysis and simulation techniques to test the following mitigation measures: (1) substitution of harmful pesticides by less harmful pesticides (simulating shift to low-risk products), (2) reducing spray drift emission to the off-crop area (part of the Dutch regulation), and (3) introducing grassy field margins to a percentage of field parcels within a landscape (a common mitigation strategy). The first two measures aim to reduce the negative impacts of applying pesticides, while the third is related to landscape management with objective of providing additional habitats for pest control agents and pollinators.

We used the spatially explicit agent-based Animal, Landscape and Man Simulation System (ALMaSS) [27] to answer the following questions: (1) what drives beetle populations in diverse agricultural landscapes of the Netherlands, (2) which mitigation measure supports beetles the best, and (3) to what extent impacts of mitigation measures are modified by the landscape context. This third point is important because if landscape structure alters the utility of a policy instrument, then it might suggest that even the national-level tailoring of EU regulation may be operating at too large scale.

## 2. Methods

The modelling environment ALMaSS is an open-source project hosted on GitLab (https://gitlab.com/ALMaSS/) with online documentation (https://projects.au.dk/almass/documentation/) using the ODdox (Overview Design doxygen) protocol [28]. The ALMaSS itself is a large system comprised of two main components, animal representations and landscape model designed to service the simulated animal. We used the carabid beetle *Bembidion lampros* (Herbst, 1784) as a model species as it represents a group of spring-breeding, carnivorous ground beetles of temperate agricultural landscapes important for natural pest control [29,30]. *B. lampros* was specifically chosen as, along with *Bembidion properans* (Stephens, 1828) which have similar life history and habitat preferences, it is widespread in the agricultural landscapes in central and northern Europe, including the Netherlands [31–33]. It has been also indicated as relevant for risk assessment of pesticides according to the EFSA Scientific Opinion addressing the state of the science on risk assessment of plant protection products for non-target arthropods [34]. In addition, the well-documented agent-based model of *B. lampros* already existed in the ALMaSS modelling framework [35].

Before carrying out simulations in ALMaSS, the specific context of the Dutch agricultural system needed to be incorporated into the modelling framework. Therefore, we divided the methods into three sections describing (1) the design of landscape model in ALMaSS and its parametrization for the Dutch landscapes; (2) the details of species model used; and (3) scenarios design and analysis for evaluating mitigation measures and landscape effects.

## 2.1 Parametrization of the ALMaSS landscape model for the Dutch landscapes

**2.1.1 General landscape model.** The highly dynamic and heterogenous character of agro-ecosystems necessitates a detailed, spatio-temporal landscape representation to provide a realistic environment for the modelled species. In ALMaSS, the spatial landscape heterogeneity is described by a detailed raster land cover map with complete coverage and spatial resolution of 1 m$^2$. Such level of accuracy is appropriate for insect species, like *Bembidion*, that utilize narrow habitats such as field margins. Each unit in the raster land cover map is classified in accordance with its landscape element type (e.g., natural or permanent grassland, field in rotation, built-up area), including detailed structures important for the species under consideration, such as hedgerows or field margins. Field boundaries are delineated, with each field belonging to a given farm unit (managed by the same farmer). Farm units are classified into different types, e.g., cattle, pig or arable farms, based on structure of crops grown and animals present on the farm.

The temporal landscape heterogeneity in ALMaSS includes both crop management throughout a year, described through individually tailored management plans for each crop, and the cropping system understood as a pluriannual crop rotation. Associated vegetation growth models for all modelled vegetation types and crops provide vegetation height, green and total biomass on a daily basis, and are driven by weather conditions (mean daily temperature, mean daily wind speed and daily sum of precipitation). Crop management plans consist of logical and ordered combinations of farm activities (related to soil cultivation, and application of fertilizers and pesticides), as well as time windows and probabilities of carrying out activities. These conditions may relate to weather, crop growth, soil type, or previous farming activities. Pluriannual crop rotations are defined separately for each farm type based on the proportion of each cultivated crop or group of crops assumed to be grown, and associated agronomic constraints (e.g. for disease prevention, or logistics). Such an approach gives a detailed dynamic landscape model with vegetation growing on a daily basis in response to the weather, and the pattern of farming activities related to each specific crop, farm, and field [27,36].

**2.1.2 'Capturing' the Dutch agricultural system *in silico*.** All details related to the generation of the landscape model for the Netherlands are described in S1 Appendix with an overview given here. This approach broadly followed [36] but required specific Dutch conditions and datasets to be taken into account.

Four main types of data were gathered and processed to obtain the necessary ALMaSS landscape inputs:

i. *Land cover / land use information.* The data were derived from the digital object-oriented topographic database of the Netherlands (TOP10NL) with level of details corresponding with topographic maps at the scale of 1:10 000 (available for browsing at https://www.pdok.nl/viewer/). We used nine classes of objects from the TOP10NL database to map 45 different layers of spatial information. Individual layers of land use / land cover information together with information on agricultural fields derived from the Agricultural Area Netherlands (Agrarisch Areaal Nederland) were then combined into a single raster landscape map in a step-by-step process.

ii. *The Land Parcel Identification System (LPIS).* LPIS records information on all agriculturally managed reference parcels (geographically delimited areas with unique identification codes) in the EU Member States, and serves as a controlling mechanism under the CAP. In the Netherlands, LPIS is managed by the Netherlands Enterprise Agency (Rijksdienst voor

Ondernemend Nederland). Information on type of crops cultivated in reference parcels, ID numbers of agricultural holdings enabling the grouping of individual reference parcels into farm units, as well as farm types were obtained for 2015 from the agricultural register 'Basisregistratie Percelen'. View and download services for the most recent agricultural register are available at thePublieke Dienstverlening Op de Kaart (PDOK) platform (https://www.pdok.nl/).

We simplified the original farm classification from the Netherlands Enterprise Agency including 38 different farm types (based on hierarchical cluster analysis taking into account similarities in acreage of different crop types, processed in R 3.4.2) to 15 farm types: animal grazing, arable grazing, arable vegetable, vegetable, greenhouse vegetable, wheat, starch potato, fruit, flower, poultry, pig and calf, horse and sheep, permanent crop, vineyards and other farms (Table A4 in S1 Appendix).

iii. *Up-to-date crop management plans.* These plans for dominant Dutch crops (winter wheat, spring barley, maize, beet, cabbage, carrot, potatoes, tulips, and managed grassland–permanent and in rotation with other crops) including time windows and probabilities of occurrence of main soil cultivation practices, as well as fertilizer and pesticide applications (together with information about the product used and its dosage) were obtained from farmer advisors from the Delphy company. To allow ALMaSS landscape simulator to modify soil cultivation depending on soil type, dominant soil type for each field parcel was defined based on the map of soil types (Grondsoortenkaart) derived from the soil map of 1:50,000 (view and download services available at https://data.overheid.nl/).

Based on proportions of crops cultivated by farms of different types, crop rotation schemes were prepared for each farm type individually (Table A5 in S1 Appendix). The crop rotation scheme consists of 100 crop entries with multiple entries of each crop type (1 crop for each 1% by area). The order of crops followed typical agronomic practices and issues such as late harvest leading to impossible sowing conditions were controlled by the built-in ALMaSS farm code. In addition, catch crops (i.e., crops grown between successive plantings of main crops) were included in the rotations following the specific rules defined by the farmer advisors from Delphy. At the start of each simulation, a random crop in the rotation is taken as the starting point for each arable field in a given farm and the next crop in the list is assumed to be grown in the same field in the following year. After 100 simulation years, all fields of one farm type would have raised each of the 100 crops in the rotation list once on each field, but in a different order. If a specific crop, e.g., maize for silage, occurs 13 times out of 100 in the rotation, it will on average occur on 13% of all fields covered by that rotation at any point in time.

All handling and analysis of spatial data were done using Python 2.7 and the Python library arcpy to access ArcGIS features, or directly in ArcGIS 10.4 [37]. The entire process of producing Dutch landscape models for ALMaSS has been programmed in Python and R scripts.

**2.1.3 Representative study areas.**   As all procedures for generating Dutch landscape models for ALMaSS are automated or semi-automated, any landscape in the Netherlands can be 'captured' and used for simulation. However, since running the CPU intensive simulations for the entire country is not feasible, we selected 10 study areas of 10 x 10 km representing a gradient from heterogeneous (small-scale) to rather homogeneous landscapes with different farm management and soil properties (Fig 1; see **in S2 Appendix** for a visual overview of study areas).

Structural and farmland heterogeneity of selected study areas were described through a set of landscape- and class-level metrics calculated at 1-m resolution using the software package FRAGSTATS v4 [38] (Table 1 and Table B1 in S2 Appendix). While some study areas were more fine-grained in structure with a high coverage of semi-natural habitats (> 15%), high

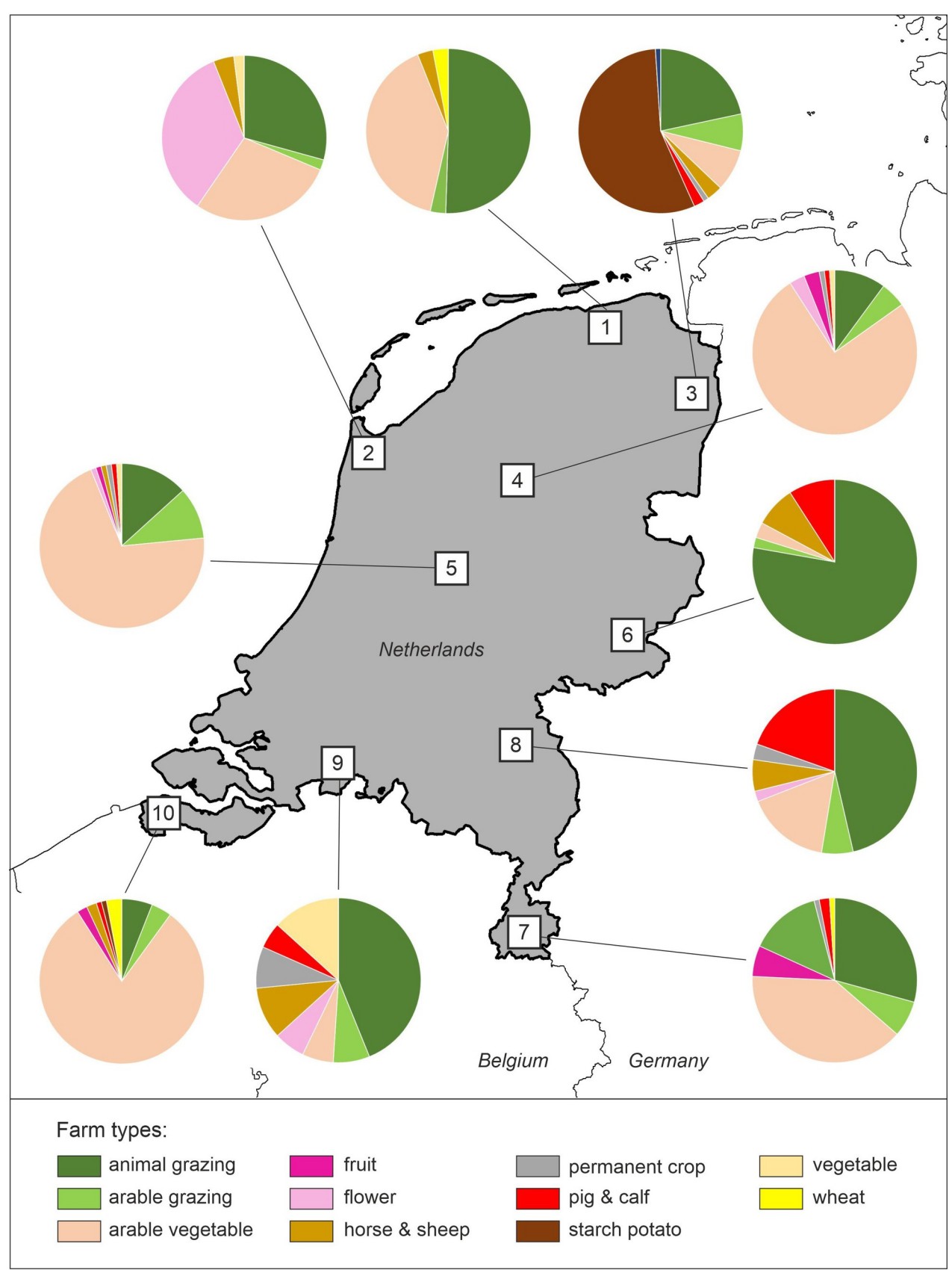

**Fig 1. Location and farming structure of the study areas.**

number of small fields (and therefore high field margins density) and high farming diversity (study areas 7, 8 and 9; Fig 2), others exhibited little structural and/or farming diversity, being dominated by large fields (mean field size > 4 ha) managed in a similar way (study areas 4 and 5 situated in the Dutch polder regions with predominantly arable farming; Figs 1 and 2).

## 2.2 Animal model

**2.2.1 Species biology and ecology.** *B. lampros* is a small (~3–4 mm), univoltine, polyphagous predator [39], common in arable farmlands. Most of individuals move by walking, and only a minority (less than 10%) exhibit functional wings and/or functional flight musculature [40]. *B. lampros* overwinters as an adult in aggregations, in the vegetated field boundaries and hedges [31]. The end of hibernation and movement from overwintering sites into field interiors is triggered by increase in air temperatures on early spring [41]. *B. lampros* can find food and reproduce in various crop types, including cereals, maize, alfalfa, beans, potatoes, oilseed rape and grasslands/pastures [42,43]. It feeds on Collembola, Diptera, mites, earthworms and small beetles [39]. *B. lampros* plays an essential role in cabbage root fly eggs and aphids pest control. Being one of the earliest spring-moving carabids, it predates on aphids when their densities are still low, which is crucial for their successful biological control [44,45].

Reproduction of *B. lampros* occurs inside fields and along its edges, from late spring up to mid-summer [46]. The species has three larval instars and one pupal stage. Development of immature stages takes place beneath the soil surface and is temperature-dependent [47,48]. The mortality in the juvenile stage is high and therefore an important factor in the population dynamics [49]. The new generation of adults appear from late summer up to early autumn, and starts hibernation on late autumn [46]. Among important external factors influencing

**Table 1. Metrics used to characterize heterogeneity of studied landscapes.**

| Type | Metric | Explanation | Range (min–max) |
|---|---|---|---|
| Landscape heterogeneity | Coverage of herbaceous habitats | % of herbaceous (without managed grasslands) in a landscape. | 4.57–11.70 [%] |
| | Coverage of woodland habitats | % of woodland habitats in a landscape. | 1.33–12.46 [%] |
| | Landscape diversity | Shannon's diversity index ($\geq 0$, without limit) of landscape element types including six categories: arable land, herbaceous semi-natural habitats, woodland (woody semi-natural habitats), build-up areas, water and others. Shannon's diversity index = 0 when landscape contains only 1 patch (i.e., no diversity), and increases as the number of different patch types increases and/or proportional distribution of area among patch types becomes more equitable. | 0.71–1.35 |
| | Landscape shape index | Normalized ratio of edge (i.e., patch perimeters) to area (class or landscape) in which the total length of edge is compared to a landscape with a standard shape (square) of the same size and without any internal edge. Values greater than one indicate increasing levels of internal edge and corresponding decreasing aggregation of patch types. | 93.58–170.92 |
| Farmland heterogeneity | Farming diversity | Shannon diversity index calculated for fields categorized into farm types (see Table A4 in S1 Appendix). | 0.81–1.81 |
| | Share of animal farms | % of farms in a landscape with >70% of grassland (permanent and perennial grassland) in the production profile (see Table A4 and Table A5 in S1 Appendix). | 8.02–85.16 [%] |
| | Number of fields | Number of fields in a landscape. | 1195–3718 |
| | Mean field size | Mean field size. | 1.40–6.65 [ha] |
| | Field boundaries density | Perimeter of field boundaries to total cultivated area. | 0.025–0.048 [1/m] |

A)

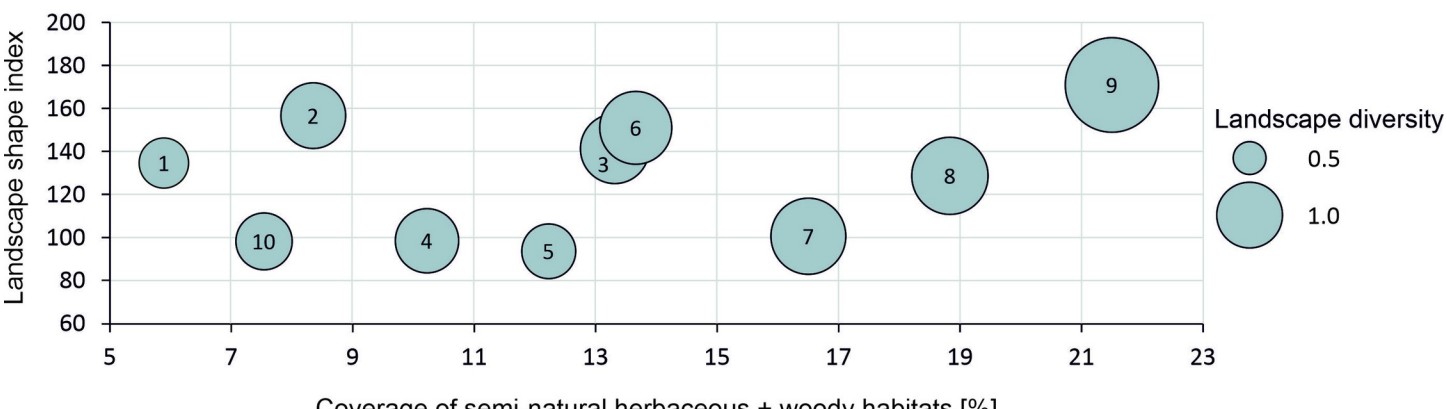

B)

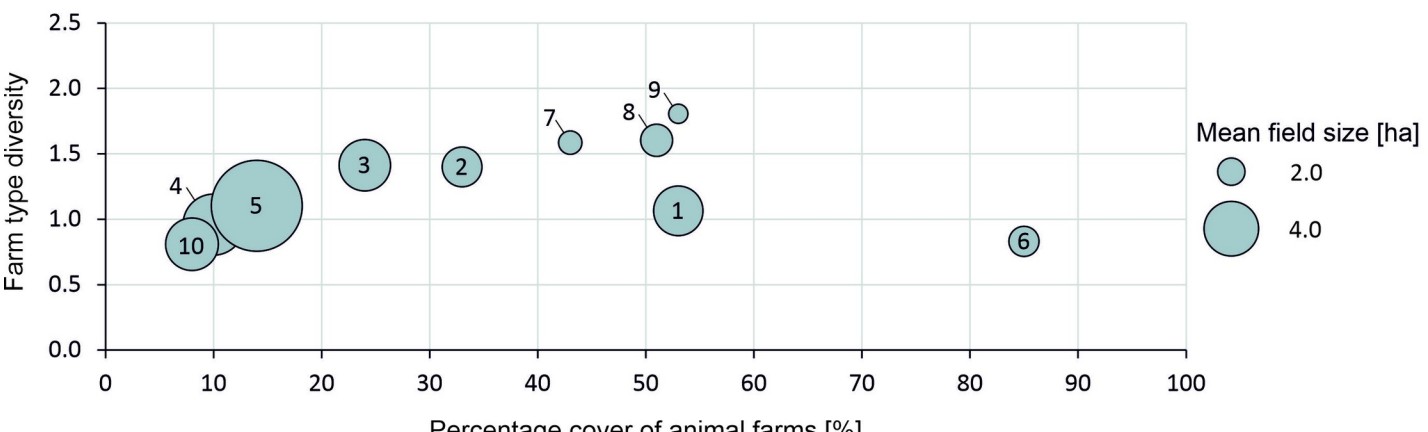

**Fig 2.** Structural (A) and farming (B) heterogeneity in studied test areas. Numbers in circles indicate study areas as shown in *Fig 1*.

adult beetle mortality are farm operations (mainly soil cultivation) [39], as well as temperatures and conditions during overwintering [50,51].

**2.2.2 Model procedures.** The original ALMaSS model for the *B. lampros* and its parametrization was described in details in [35] as well as in model's ODdox documentation (https://www2.dmu.dk/ALMaSS/ODdox/Bembidion/index.html), therefore only summary is provided below. As in the real world the species occurs in high densities, the model uses the super-individual concept, which means that each beetle agent in the model represents 100 beetles in the real world [22]. *Bembidion* behaviour is simulated on a daily time-step. All individuals are categorized as being members of four life-stages: egg, larvae, pupae, and adult female. We assume that all the adult females are being fertilized and can produce eggs, hence adult males are not being followed in the simulation.

Movement in the model is determined by four parameters: (1) a directional vector that indicates the preferred direction, (2) a weight indicating the strength of the bias towards the directional vector, (3) a maximum allowed distance per time step, and (4) the probability of a beetle accepting a sub-optimal habitat [27]. Within or in-between habitats of similar quality (e.g.,

dispersal within fields during reproduction or foraging), movement is determined by random choice. Autumn migration (directional movement) towards hibernation sites is determined by a probability distribution starting on 1st of October, while spring dispersal (directional movement) towards foraging sites is triggered by temperature.

Primary drivers in the model are temperature-controlled developmental rates of eggs, larvae and pupae (defined based on [47,48]), together with adult female interactions with the landscape. At high densities species model invoke density dependence via intraspecific predation. This means that the number of beetles that could be present within a 3-m x 3-m square surrounding any beetle is limited to two adults and two larvae (assuming that the resolution of the landscape model is 1m$^2$). Additional beetles are removed from the simulation. Beetle mortality caused by farm operations are defined based on [52], while mortality during overwintering based on [50]).

The response to the pesticide is included in the model by assuming a threshold environmental concentration above which there is a daily probability of mortality (*p*). This probability is calculated from the following equation:

$$(1 - m) = (1 - p)^d,$$  [Eq 1]

where *m* is the proportion of beetles assumed to die (e.g. 0.9 for 90% mortality over the test period) and *d* is the number of days over which the test was carried out. If the beetle finds itself in a 1-m$^2$ grid cell with an environmental concentration above the trigger, then it is assumed to die with the probability *p*. There is no dose–response, so the maximum death rate is set as *m* over *d* days.

## 2.3 Simulation scenarios

We implemented the following mitigation measures in ALMaSS (summary provided in Fig 3): (i) substitution of harmful pesticides by less harmful pesticides; (ii) reduction of spray drift emission to the off-crop area; and (iii) introduction of grassy field margins to a certain percentage of fields within a landscape. Measure (i) aims to provide a direct reduction in emission and/or impact of pesticides on the fields, while measure (ii) and (iii) are used to reduce the emission to off-crop areas, such as field ditches. In addition, measure (iii) provides additional overwintering, reproduction or foraging habitats for modelled species.

### 2.3.1 Pesticide-related mitigation measures

*Toxicity*. We used realistic pesticide application scenarios provided by extension advisors (see section 2.1.2). However, to reduce computation time, we classified the pesticides into three groups based on toxicity instead of values related to specific active ingredients and/or products to account for broad diversity of products used in the Netherlands. For insecticides, we assumed high, medium or low toxicity, and for normal fungicide and herbicide application we assumed no direct impact on the carabid beetles. We defined the high toxicity level of insecticides as related to beetle field lethality rate of 90% (LR90) measured for a foliar insecticide-spray application over seven days, which gives daily probability of mortality *p* of 0.28. For a subset of scenarios, decreased field lethality to medium (LR50 corresponding with *p* = 0.09) or low (LR25 corresponding with *p* = 0.04) level was assumed; Fig 3). The decreasing field lethality simulates substitution of harmful insecticides by less harmful insecticides but can be also considered equivalent to a reduction in exposure or application rate, or toxicological impact of insecticides in the field. It can therefore also simulate the impacts of canopy cover in the sense that if canopy is more dense then less insecticides reach the beetles moving on the ground meaning that the probability of field lethality will decrease.

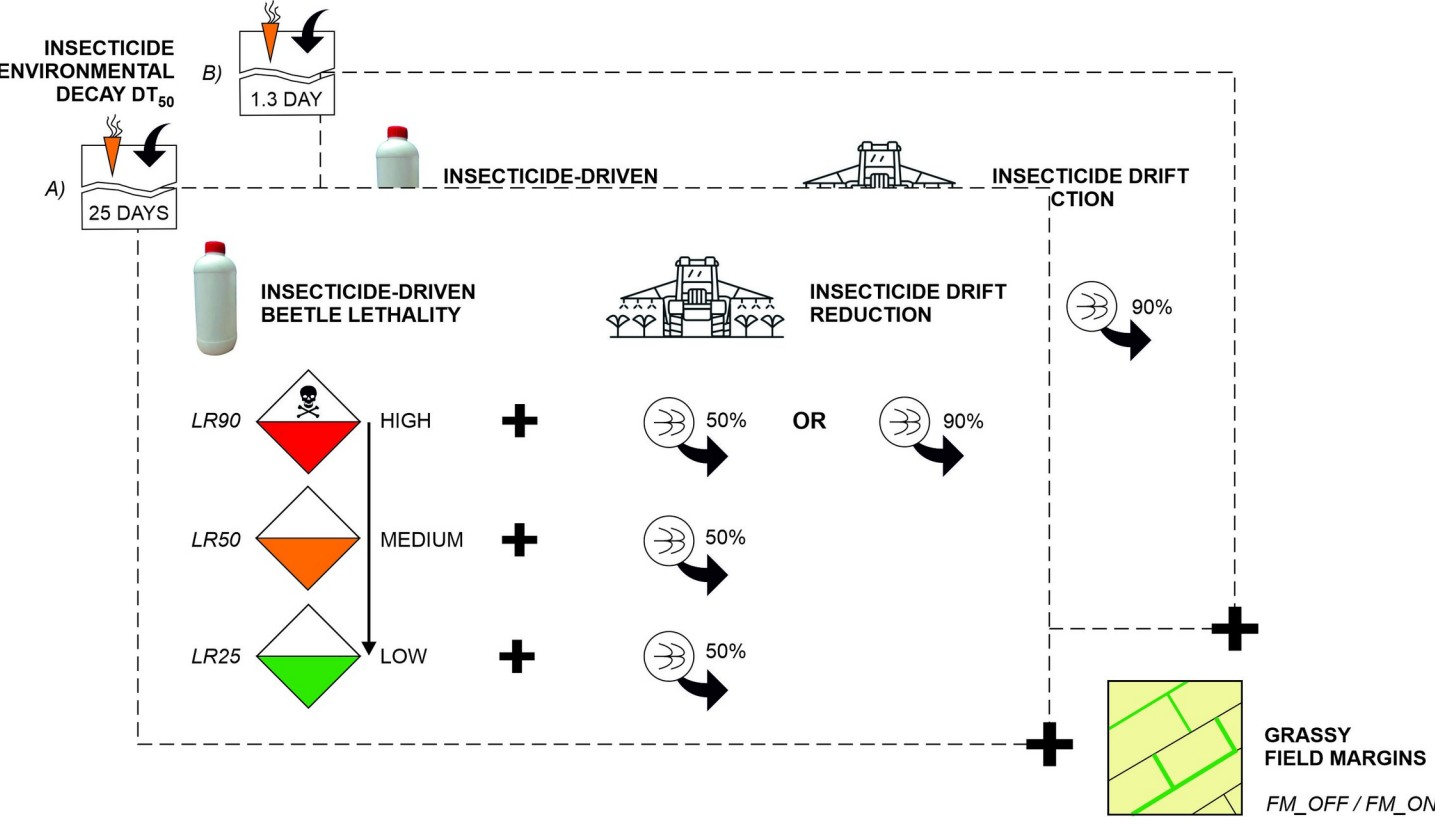

**Fig 3. Summary of implemented mitigation measures.** Three levels of insecticide toxicity defined in relation to insecticide-driven beetle field lethality rates, high (LR90), medium (LR50) and low (LR25), were tested in combination with drift reduction of 50% or 90% (only for scenario LR90) and additional grassy field margins not included (FM_OFF) or included (FM_ON) in a landscape. Eight combinations of these scenarios were tested for insecticides with two different environmental decay values: (A) $DT_{50}$ of 25 days, and (B) $DT_{50}$ of 1.3 day. These 16 combinations of scenarios were evaluated in ten study (Fig 1).

*Environmental decay.* In simulations, we considered two significantly different values for environmental degradation time ($DT_{50}$) at 20˚C: 1.3 days and 25 days (Fig 3). These values represent two different groups of insecticides with high and low environmental degradation properties and were chosen based on measured $DT_{50}$ values of insecticides most commonly used in the Netherlands. The temperature dependence of $DT_{50}$ value was defined according to the following equation [53]:

$$DT_{50}(T1) = DT_{50}(T2) \times exp\left(lnQ_{10} \times \frac{T2 - T1}{\Delta T}\right) \quad\quad [Eq\ 2]$$

where $DT_{50}$ (*T2*) is a half-life at experimental temperature *T2* (in days), $DT_{50}$ (*T1*) is half-life at temperature of interest *T1* in days, *ΔT* is equal to 10˚C, and $Q_{10}$ is the ratio of pesticide degradation rate coefficients ($k_2/k_1$) at a temperature *T1* that is 10˚C lower than a temperature *T2*. *T2* was set to 20˚C because it is the reference temperature for measuring degradation rates, and $Q_{10}$ was set to 2.58 as suggested by [53].

To allow direct comparisons of the toxicity-levels applied when considering different $DT_{50}$, a toxic unit approach was used. To ensure that beetles could be exposed above the trigger threshold for at least the period defined by $DT_{50}$ (1.3 or 25 days), a treatment rate of 41.78 of the trigger concentration at each LR was used for insecticides with $DT_{50}$ = 1.3 days, and a treatment rate of 1.22 of the trigger concentration at each LR was used for insecticides with $DT_{50}$ =

25 days. These rates (41.78 and 1.22) were calculated as follows:

$$treatment\ rate = 1 / [0.5^{1/DT50\ [days]} \times duration\ of\ effect\ [days]] \qquad [Eq\ 3]$$

This means that for low $DT_{50}$ the probability of dying if exposed in the first day is very high whereas for high $DT_{50}$ the daily chance is lower but effect period longer.

*Spray drift*. During the application of pesticides, part of the spray liquid may be carried out of the treated area by wind or the air stream of the air-blast sprayer and may be further spread by drift and result in a contamination of adjacent crops and waters. In all simulations we assumed that drift occurs up to 12 m from the edge of any sprayed field, following the equation by Rautmann et al. [54]:

$$\%\ drift = A \times dist^B \qquad [Eq\ 4]$$

where *A* and *B* are coefficients that depend on crop and *dist* is the distance between the field border and a point downwind the field (m). For field crops assuming downward spraying, and no drift reducing technologies, A = 2.7705 and B = -0.9787 [54]. In the Netherlands, farmers must reduce the spray drift by at least 50%, and the newest requirements (from January 1, 2018) refer to 75%. However, a survey amongst 600 farmers showed that 70% of Dutch farmers reduced the spray drift by 90% [18,55]. This was partly voluntarily, but mandatory when they used toxic pesticides for which there is an additional restriction to reduce the spray drift by at least 90%. In our study, we used two scenario's for spray drift, i.e. the baseline with 50% spray drift reduction and a scenario in which all farmers reduce the spray drift by 90%. The latter is the aim of the new policy document "Vision on crop protection 2030" [56]. We re-calculated the coefficients of [Eq 4], for 50% and 90% drift reduction and tested their influence in a subset of scenarios for the ground beetles (Fig 3).

**2.3.2 Landscape-related mitigation measures.** According to Dutch legal requirements, a crop-free and spray-free buffer zone is to be maintained alongside surface water. The width of the buffer zone ranges from 0.5 m to 1.5 m, depending on crop type. Because the resolution of ALMaSS is 1 m, the crop-free buffer zone was set to 1 m for all crops. We assumed that these crop-free zones are fallow at the beginning of each simulation year with grass re-growing during the year.

To test different landscape management strategies, a land-cover map of each study area was used in two artificially manipulated forms. The first included fields that were completely covered by the crop, the second was to create a field boundary in-field around the crops (not subjected to the same agricultural practice as the crop itself such as ploughing or harvesting or pesticide application). Grassy boundaries of 4-m width were applied to 50% of fields in each study area, in a subset of scenarios (Fig 3).

**2.3.3 Simulation set-up.** The simulation for each combination (16 combinations of scenarios, see Fig 3) was replicated 10 times, as this number of replicates was shown to be sufficient to account for between-replicates variability, which in case of the *Bembidion* model is generally very low [22]. Initial conditions for each replicate differed in the distribution of beetles across the study area and the initial allocation of crops in the fields, but not in the starting number of super-individuals which was set to 200 000 per 100 $km^2$. The starting number of super-individuals did not determine the final outcome of model simulations as after a few years of simulation runs beetle populations approached densities that were independent of the initial population size. All simulations were run for 30 simulation years with no burn-in period (see ALMaSS configuration files, S3 Appendix).

Weather conditions were selected to represent the 1998–2017 period from De Bilt climatic station, which is considered representative for Dutch conditions (data from European Climate Assessment & Dataset; www.ecad.eu).

**2.3.4 Simulation data extraction and analysis.** From each simulation, three endpoints were analysed: overall beetle population density (i.e., total number of adult female beetles divided by the landscape area, i.e. 100 km$^2$), Occupancy (i.e., beetle distribution defined as proportion of grid cells in the landscape with at least 100 adult female beetles) and Abundance (mean density of adult females in the occupied areas). The latter two endpoints are presented as Abundance-Occupancy Relationship (AOR) plots [57]. Although the spatial resolution of landscape model in ALMaSS is 1 m$^2$, for Occupancy and Abundance calculation grid cells of 50 m$^2$ were used. Overall beetle population density, and beetle Abundance and Occupancy were measured at day 59 of each year (March 1st) of each simulation, and then means over 30 years of the simulations were calculated. These endpoints were then averaged across 10 replicate runs for each scenario. To determine the effect size of each mitigation measure on *B. lampros* population, the impact of each scenario relative to the baseline was used and compared over time. To highlight improvements related to the treatments, we set baseline conditions according to the 'worst-case' scenario with high insecticide toxicity (LR90), insecticide drift reduction of 50% and no additional field margins added to the landscape (FM_OFF). Two such baselines were created, one for low and high (1.3 and 25 days) environmental half-life, respectively.

The influence of metrics describing landscape and farmland heterogeneity (Table 1) on mean overall beetle density, abundance and occupancy in the baseline ('worst-case') scenarios A and B, and on changes of mean overall beetle density, abundance and occupancy in response to applied mitigation measures was tested with multiple regression models. Metrics of landscape and farmland heterogeneity were first checked for correlations and those with Pearson correlation coefficient $> |0.7|$ were removed from the analysis. As a result, the following metrics were used as explanatory variables in the regression models: landscape shape index, landscape diversity, farming diversity, percentage of animal farms and number of fields in a landscape. Separate multiple regression models were constructed for pesticide-related and landscape-related mitigation measures, for each of the endpoints considered. The best performing models were chosen with backward stepwise algorithm based on the Akaike information criterion. For each model, we reported adjusted *R-squared* value and *p*-value. Models and variables with p-value $\leq 0.05$ were considered significant. Calculations were performed in R environment [58] with 'stats' and 'MASS' packages [59].

# 3. Results

This section first describes the beetle populations simulated under the baseline scenarios. Then, the effectiveness of different mitigation measures on the population of *B. lampros* in 10 study areas is presented. For each scenario, coefficients of variation (cv) for the simulation endpoints were computed from 10 replicate runs and were in all cases not greater than 2.3% for the mean overall population density, 1.0% for the mean abundance and 1.4% for the mean occupancy. Simulation replicates were therefore very similar and no more than 10 replicates were needed.

## 3.1 Beetle populations under the baseline scenarios

The beetle populations under the baseline scenarios (LR90 combined with spray drift reduction of 50%, and no additional field margins) varied across study areas (Fig 4). Beetle populations were slightly lower when a pesticide with a longer half-life was applied, although the

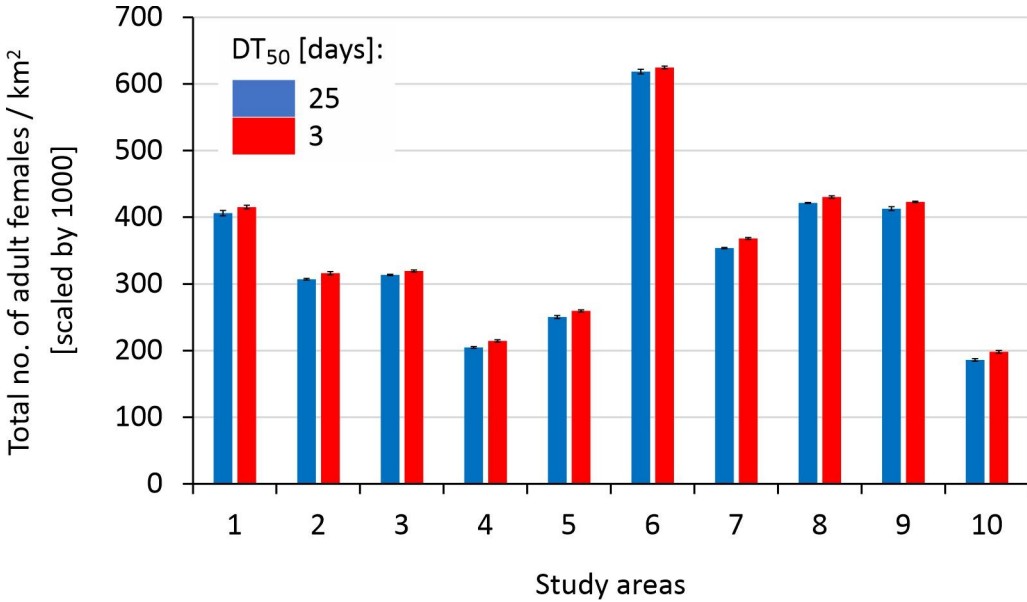

**Fig 4.** The mean (+/- SD) overall population density of B. lampros in 10 study areas (as shown in Fig 1) according to the baseline ('worst-case') scenario (LR of 90%, insecticide drift reduction of 50%, and no additional field margins added to the landscape) with $DT_{50}$ of 25 days (baseline scenario A; blue bars), and $DT_{50}$ of 1.3 days (baseline scenario B; red bars).

difference was very small (Fig 4). The mean overall beetle density and Occupancy in baseline scenario A increased with the landscape diversity ($p < 0.01$) and percentage of animal farms in a landscape (i.e., with the amount of managed grasslands; $p < 0.01$; Table D1 and D3 in S4 Appendix). Mean beetle Abundance was influenced only by farmland heterogeneity. It increased with the area of animal farms in a landscape ($p < 0.01$), but decreased with increasing number of fields (and decreasing field size) in a landscape; $p = 0.03$; Table D2 in S4 Appendix). Similar trends were observed when the $DT_{50}$ was decreased to 1.3 day (baseline scenario B; see Tables D1-D3 in S4 Appendix).

### 3.2 Effects of mitigation measures on beetles

The impacts of applying each mitigation measure separately are described below. The results presented in sections 3.2.1–3.2.3 refer first to scenarios in which $DT_{50}$ was set to 25 days, and then comparison to scenarios with $DT_{50}$ of 1.3 days is described. The combined effects are shown in section 3.2.4.

**3.2.1 Decreasing the toxicity of applied insecticides.** Decreasing the toxicity of applied insecticides had positive impact on beetle populations in all analysed study areas, but its magnitude varied considerably and was substantially influenced by landscape and farmland heterogeneity. In all scenarios that reduced insecticide toxicity, both the mean overall beetle density and Occupancy increased substantially relative to the baseline (Fig 5). However, this was not the case for the mean beetle Abundance, which in some study areas was almost stable or even somewhat decreased (Fig 5) due to increased Occupancy. The magnitude of observed impacts was higher in scenarios with $DT_{50}$ of 1.3 days (Fig 5B) than 25 days (Fig 5A).

In scenarios with $DT_{50}$ of 25 days, decreasing insecticide toxicity from high (LR90) to medium (LR50) increased the mean overall beetle density up to 11%, and mean Occupancy up to 10%. Further reduction of insecticide toxicity to LR25 resulted in relative increases exceeding 20% and 15% for the mean overall beetle density and Occupancy, respectively. The relative

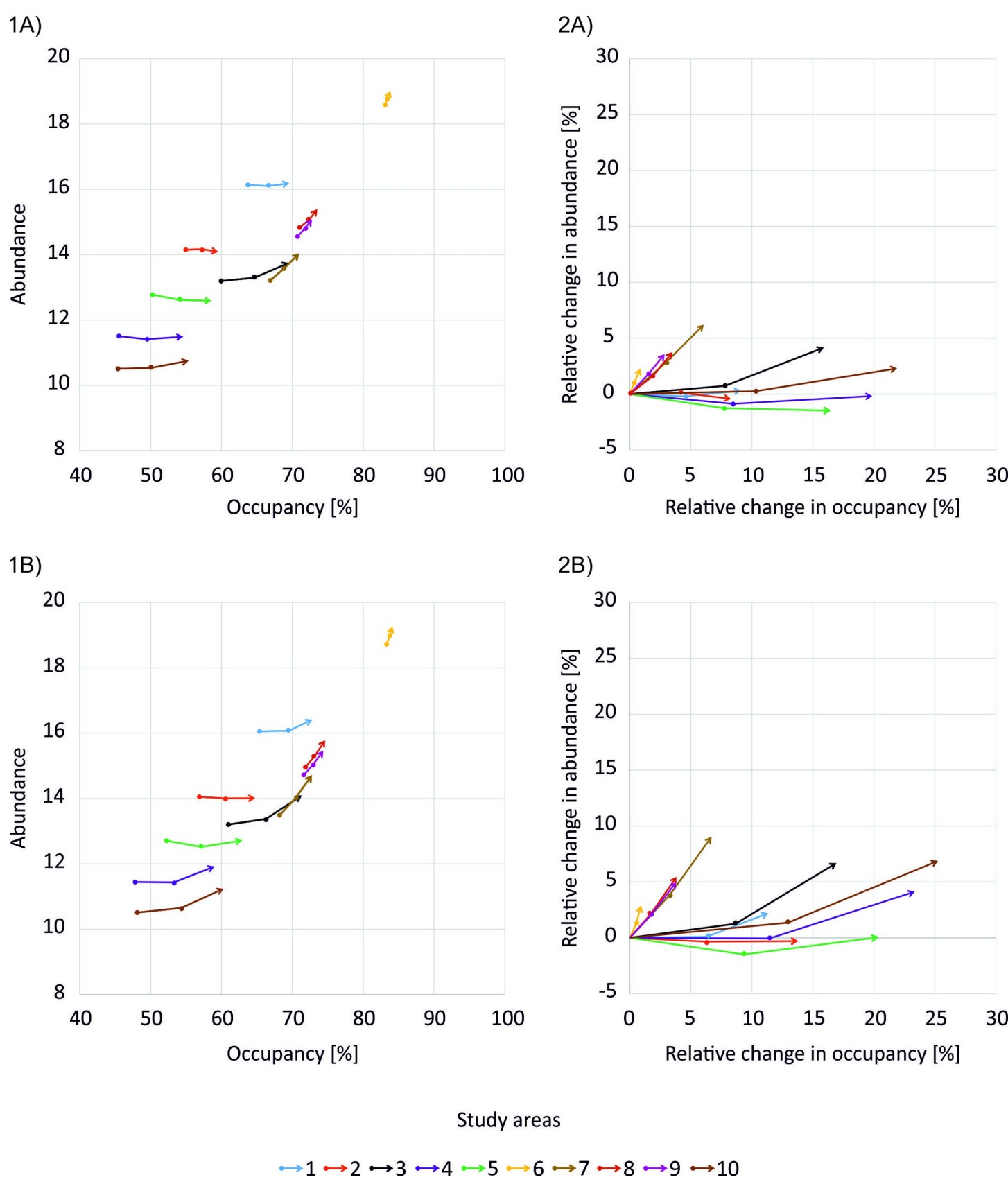

Study areas

→1 →2 →3 →4 →5 →6 →7 →8 →9 →10

**Fig 5.** Changes in mean beetle occupancy and abundance of B. lampros populations (plot of Abundance to Occupancy Relationship, AOR) in scenarios decreasing the insecticide toxicity and with $DT_{50}$ of (A) 25 days or (B) 1.3 days: (1) absolute changes, (2) relative changes in comparison to the baseline scenario (with LR of 90%, drift reduction of 50%, and no additional field margins added). Dots mark consecutively, starting from the bottom, the baseline scenario (0, 0), and changes in AOR resulted from decreasing the insecticide toxicity to LR50, and LR25.

change in the mean overall beetle density and Occupancy decreased with the area of animal farms in a landscape ($p < 0.01$; Table D4 and D6 in S4 Appendix). The study areas most positively impacted were those with the lowest baseline beetle populations, i.e., those with the lowest area of managed grasslands in a landscape and, at the same time, high proportion of arable vegetable farms (study areas 4, 5 and 10). In addition, a high increase in mean overall beetle density and Occupancy was found in the study area 3 dominated by potato farms. Mean beetle Abundance was impacted more substantially only when insecticide toxicity was decreased to LR25, but still for study areas 4 and 5 the Abundance had not reached the baseline values (Fig 5A). The relative change in the mean beetle Abundance increased with the number of fields in a landscape ($p = 0.02$; Table D5 in S4 Appendix).

The relative increase in mean beetle Occupancy in response to decreasing insecticide toxicity was up to 5% higher in scenarios with $DT_{50}$ of 1.3 days (Fig 5B) than 25 days (Fig 5A), while for beetle Abundance the difference was up to + 3% (Fig 5B). In scenarios with $DT_{50}$ of 1.3 days, the relative change in mean overall beetle density and Occupancy not only decreased with the area of animal farms in a landscape ($p < 0.01$ for both beetle density and Occupancy) as in scenarios with $DT_{50}$ of 25 days, but also with the farm diversity ($p = 0.02$ for beetle density, and p = 0.03 for beetle occupancy). In addition, in scenarios with $DT_{50}$ of 1.3 days relative change in mean beetle Occupancy was negatively impacted by the landscape diversity ($p = 0.01$; Table D4 and D6 in S4 Appendix). The regression model for the relative change in mean beetle Abundance was non-significant (Table D5 in S4 Appendix).

**3.2.2 Increasing the level of drift reduction.** No effect of spray drift reduction was detectable when insecticides were applied with low application rate (1.22) corresponding with $DT_{50}$ of 25 days. When high application rate of insecticides (41.78) corresponding with $DT_{50}$ of 1.3 days was used, effects of spray drift reduction on mortality were visible; however, its effect on beetle mortality was already very small with drift reduction of 50%. Therefore, increasing the reduction of pesticide drift from 50% to 90% had no significant effect on beetle populations in analysed study areas (the effect was smaller than in-between replicates variation).

**3.2.3 Introducing grassy field margins.** Introducing grassy field margins of 4-m width to 50% of cultivated fields in a landscape increased beetle population size in all study areas, but the impacts were smaller in comparison to toxicity-related scenarios. In addition, the magnitude of impacts did not differ substantially between scenarios with $DT_{50}$ of 25 (Fig 6A) and 1.3 days (Fig 6B).

In scenarios with $DT_{50}$ of 25 days, the relative changes in the mean overall beetle density and Occupancy decreased with the share of animal farms in a landscape ($p = 0.02$ for beetle density, and p < 0.01 for occupancy; Table D7 and D9 in S4 Appendix). Similar trends were found in scenarios with $DT_{50}$ of 1.3 days, however the relative change in mean overall beetle density was also negatively influenced by farm diversity (p = 0.02; Table D7 in S4 Appendix), and relative change in mean beetle Occupancy increased with the number of fields in a landscape (p = 0.05; Table D9 in S4 Appendix). Introducing grassy field margins only slightly increased the mean beetle Abundance (up to 3%; Fig 6), and the regression models for the relative change in mean beetle Abundance were not significant in scenarios with $DT_{50}$ of both 25 and 1.3 days (Table D8 in S4 Appendix).

**3.2.4 Combined effects.** Application of combined mitigation measures (i.e., decreasing of insecticide toxicity and introduction of field margins) resulted in a substantial relative increase

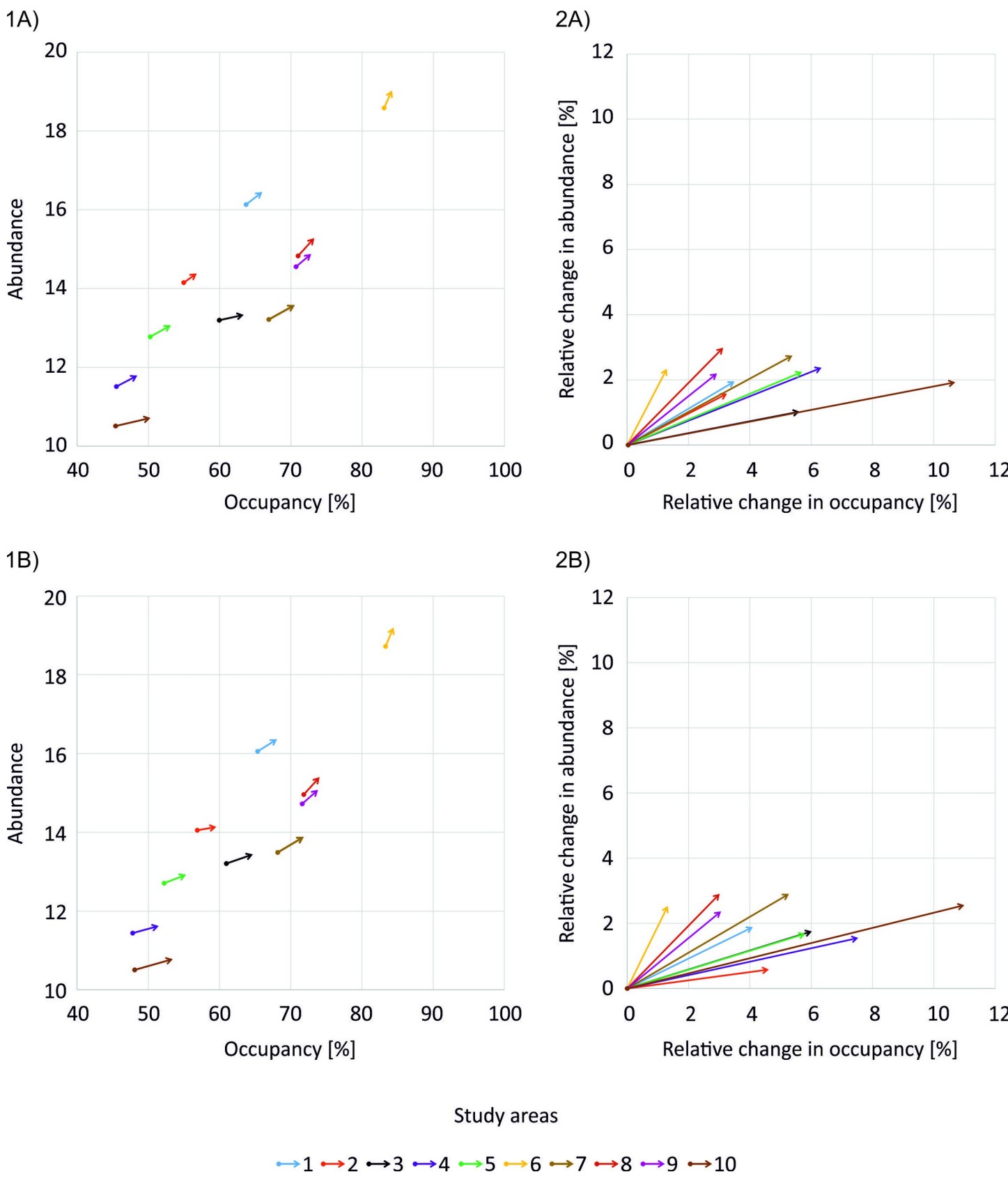

Study areas

→1 →2 →3 →4 →5 →6 →7 →8 →9 →10

**Fig 6.** Changes in mean beetle occupancy and abundance of B. lampros populations (plot of Abundance to Occupancy Relationship, AOR) in scenarios introducing additional field margins and with $DT_{50}$ of (A) 25 days or (B) 1.3 days: (1) absolute changes, (2) relative changes in comparison to the baseline scenario (with LR of 90%, drift reduction of 50%, and no additional field margins added). Dots mark consecutively, starting from the bottom, the baseline scenario (0, 0), and changes in AOR resulted from decreasing the insecticide toxicity to LR50, and LR25.

in both mean beetle Occupancy and Abundance (Fig 7). The biggest impacts were noted for study area 10 with relative increase exceeding 15% for mean beetle abundance and 35% for beetle occupancy. In scenarios where insecticides with short environmental degradation time were used, those impacts were magnified, especially for mean beetle Abundance (Fig 7).

## 4. Discussion and conclusions

Biodiversity-related agricultural polices aim at reducing negative impacts of pesticide use on farmland habitats and, at the same time, at increasing natural pest control through supporting populations of natural pest enemies. However, it has been suggested that the mitigation measures proposed in agri-environmental programs, even if tailored for a certain group of species, may not be successful if not also tailored to the ecological and spatial context of the specific agricultural landscape of interest [60]. In this study, by using a combination of high-resolution dynamic landscape models and a detailed spatially-explicit agent-based model, we were able to realistically simulate the behavior and population dynamics of a non-target arthropod species *B. lampros* in the various agricultural landscapes of the Netherlands. The subsequent analysis allowed us to determine the major drivers of beetle populations in relation to different farming systems, as well as to assess the effectiveness of selected mitigation measures under various initial conditions (including varying landscape and farmland heterogeneity). The measures tested were suggested in the document 'Healthy Growth, Sustainable Harvest' on sustainable crop protection policy in the Netherlands for the 2013–2023 period [17] but are generally relevant at the EU level.

Below we consider the major facets of the beetle responses before drawing conclusions for agricultural policy measures.

### 4.1 Factors controlling beetle population size in agricultural landscapes

Simulated beetle populations under initial conditions (i.e., before mitigation measures were applied) differed considerably between landscape and farming systems. This confirms the importance of landscape structure for carabids populations in agroecosystems [9,61–63]. Beetle populations were larger and distributed over larger areas in more diverse landscapes with a high proportion of animal farms (i.e., extensive area of managed grasslands) and, at the same time, low area coverage of arable vegetable farms. Highly diverse landscapes have higher quantity of non-arable habitats, and therefore extended arable to non-arable boundaries supporting effective colonization of arable fields by beetles [10,64]. This implies that the starting conditions for assessment of policy instruments should be considered before an assessment of effects are made. In the Dutch farming systems, managed grasslands provide large areas free of insecticide spraying and intense soil cultivation. They are thus favorable for beetles, especially if embedded in a landscape with high intensity use. In mixed farming systems "grasslands and croplands complement each other" by providing continuous and different resources to the species throughout the year [65]. It is, however, important to note that in our simulations we did not consider possible negative effects of intensive grazing on beetles. Large herbivores may impact arthropods in various ways, by directly increasing mortality rate and disturbance or, indirectly, by changing vegetation structure of grasslands and their abiotic conditions [66]. However, if managed carefully, grasslands can make an

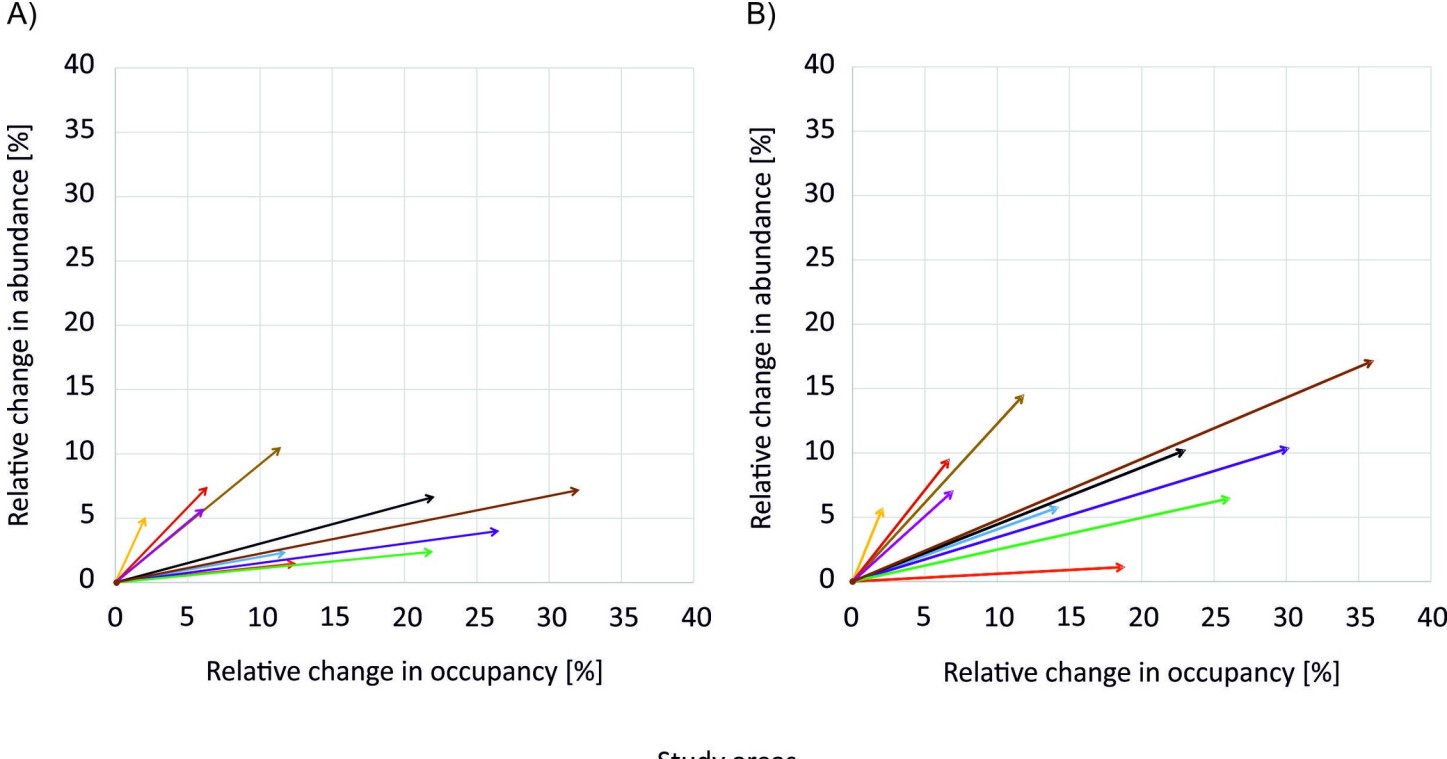

**Fig 7. Relative changes in mean beetle occupancy and abundance of B. lampros populations (plot of Abundance to Occupancy Relationship, AOR) in analysed study areas when comparing the "worst" and the "best" scenario, i.e., the toxicity is reduced from high to low, and field boundaries of 4-m width are added to 50% of fields in a landscape.** Effects showed for scenarios with $DT_{50}$ of (A) 25 days or (B) 1.3 days.

important contribution to the biodiversity of agricultural landscapes, at the same time supporting population-based regulating services (e.g., pollination and biological control) [67]. More diverse landscapes also promote other non-target arthropods, including spiders [68], and butterflies [69,70]. The amount and proximity of grassy habitats are known to strongly drive carabids communities [65,71] but also to support, in general, biodiversity in agricultural landscapes [72,73]. This suggests that policy goals could partly be achieved by promotion of mixed farming strategies.

Local beetle Abundance, unlike Occupancy, was harder to predict as it is a result of dynamics between source and sink habitats, and their distribution within the landscape. The fact that beetle density in occupied areas was negatively influenced by the number of fields in a landscape seems incongruous at first. However, the number of fields in a landscape was correlated with number of farms in our study areas, and larger number of farms could increase crop rotation and diversity. This would normally be considered a good thing, but as a result insecticide applications were more evenly distributed within the landscape (occurring in many small but scattered fields). This effectively increased the treated area:edge ratio and subsequent mortality occurring at this boundary [21]. Like flying insects that are evenly redistributed across agricultural landscapes [74], if we have an evenly distributed population of even somewhat mobile beetles, and the distribution of toxic locations are many and evenly spread, the probability of beetle exposure increases considerably.

## 4.2 Key factors in the effectiveness of mitigation measures

Here we compared effectiveness of selected measures from two treatments, pesticide- and landscape-related. Effects of drift-reducing measures at the population-level were of little importance. This is almost certainly due to the fact that at the time of spraying, the population was largely present in the field and exposure was reduced marginally by the drift-reduction. Much larger benefits resulted from reduction of insecticide toxicity, and these generally exceeded those associated with the introduction of additional field margins of 4-m width to 50% of cultivated fields (Fig 8). Such effects were expected, as already indicated by other ALMaSS simulation studies [22,26]. The magnitude of these effects and the interplay between beetle Abundance and Occupancy were, however, strongly driven by the landscape context. Depending on the starting context (landscape and farmland heterogeneity), the results of the mitigation measures may predominantly increase the range of the beetles or increase their local abundance, or influence both strongly. For a high initial beetle Occupancy (> 70%), decreasing insecticide toxicity had minimal impacts on increased Occupancy because insecticide pressure on beetles was low even before the mitigation measure were applied, and

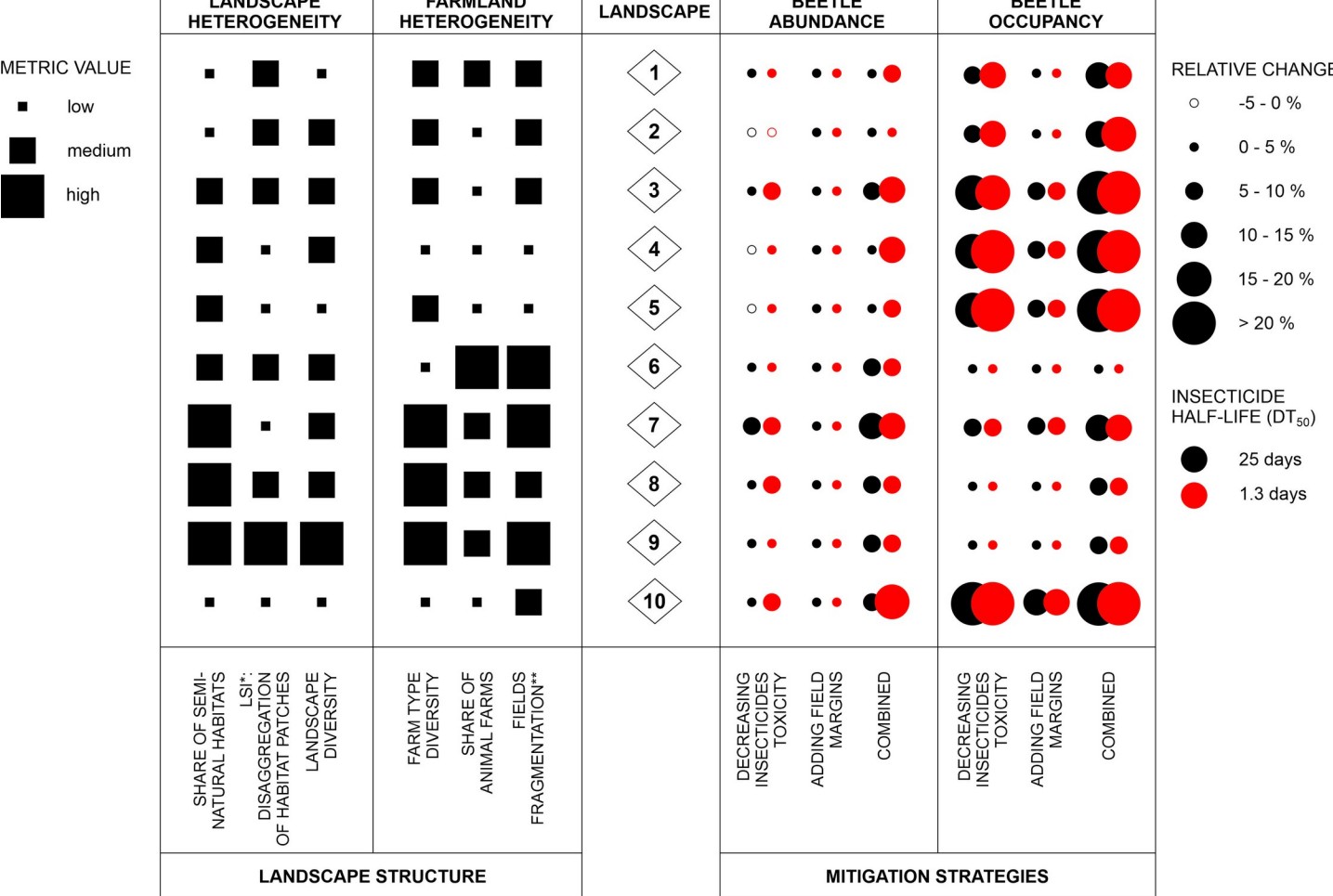

**Fig 8. Relative impacts of analysed mitigation strategies on Bembidion lampros populations in relation to landscape structure.** Mitigation strategy of increasing reduction of drift was omitted due to non-significant effect. LSI stands for landscape shape index and is a proxy of disaggregation of habitat patches. Fields fragmentation was defined as a measure combining number of fields, mean size of fields and field boundaries density (for more explanation see Tables B2-B3 in S2 Appendix).

populations were limited by other factors. However, Abundance could increase in the same scenario. On the other hand, in areas with low initial beetle Occupancy, decreasing insecticidal toxicity permitted more rapid colonization of new areas but with low Abundance. In such study areas, a substantial increase in beetle Abundance (> 5%) could only be achieved when pesticide- and landscape-related measures were combined (Fig 8). This confirms that in mixed farming systems field margins are important rather as a supporting and not stand-alone mitigation measure, as they generally improve the effects of a reduction in insecticide toxicity [26].

The impacts of mitigation measures analyzed were modified by the insecticide's half-life ($DT_{50}$). In general, in scenarios with a low $DT_{50}$ of 1.3 days, positive effects of toxicity-related mitigation measures were higher compared to long-persistence insecticides (Fig 8). Pesticides with shorter half-lives have shorter periods during which toxic levels are exceeded, and therefore allow beetle populations to recover more easily. Because of the toxic-unit approach used, initial mortality rates were not very different for high and low persistence pesticide, but the period of population influence was. In agreement with the general assumptions, this may suggest shifting towards low persistent insecticides would be beneficial, but only if this did not result in an increase in number of treatments, which was not considered in this study.

## 4.3 Impacts of model beetle dispersal abilities on simulation results

The observed beetle population dynamics was largely driven by the model beetle dispersal abilities and habitat preferences. In this study we allowed beetle to move with intermediate maximum dispersal rate of 14 m per day [35] within preferable habitats. This is a simplified approach as dispersal rate may vary depending on the habitat quality [75] and between reproductive and non-reproductive periods [76].

Previous simulation studies showed that increasing dispersal ability results in larger beetle populations [21,35] and suggested that there is a minimal movement rate essential for beetle long-term survival [35]. In general, larger movement range increases the probability of reaching the favorable habitat and thus positively impacts the population growth. For lower dispersal abilities the magnitude of this effect increases with landscape and farmland heterogeneity, but higher model beetle dispersal rates (> 20 m per day) lead to disassociation of the population dynamics from landscape structure [35]. In addition, it has been shown [21] that when beetles were able to disperse on longer distances, they were in general more impacted when insecticides were sprayed. The magnitude of these impacts, however, varied spatially and depended on spatial distribution of pesticide application areas. This highlights the interaction between beetle dispersal abilities, habitat quality and pesticide effect, and shows that the impacts on beetle populations being observed are highly context-specific.

## 4.4 Consequences for agro-policy

The important lesson from this study is that population responses to applied mitigation measures can be very complex and depending on a landscape context. The decision should be made in relation to farming system, policy aim, and landscape structure. This would suggest the need for implementation of locally-adapted agro-polices. Nevertheless, some general recommendations can be made based on our results.

**Shift towards 'low-risk' products as a priority.**   In general, decreasing insecticide toxicity was more effective in supporting beetle populations. Promoting shift towards 'low-risk' plant protection products, proposed in the Duch policy, should be the foremost priority, especially in areas dominated by intensively sprayed crop and plant species, such as potatoes or flower bulbs.

**Drift reduction not a useful measure for promoting in-field ecosystem services.**   Perhaps obvious, but drift reduction is targetted at species living in the surround habitats, not

those that live in the field itself at the time of spraying. This result does not mean that drift reduction is not a good thing, but it needs to be considered in the context of the policy goals.

**Mitigation measure should reflect management goal.** The simulations showed that mitigation strategies tested individually (especially decreasing toxicity of applied insecticides) were more effective at supporting increase in beetle's Occupancy than Abundance in study areas dominated by rotational farming system. This is a desirable outcome, as by increasing beetle Occupancy we are increasing the number of fields subject to service, rather than focusing on improvement of pest control only in a few, selected fields. In an agricultural system dominated by permanent crops with coexistence between beetle distribution and crops, abundance increases may be more desirable. Such management goals require the implementation of combined mitigation strategies, which would not only decrease the pesticide pressure but also increase the amount of source habitats in a landscape. However, once again, we should consider the management goal as here we assumed pest control, but if it was conservation then an increase in Occupancy is likely to be more highly favored.

**The effectiveness of mitigation measures are modified by the spatial dynamic factors.** This means that, although general effects could be predictable, the actual magnitude of these effects is highly variable and modified by various spatial dynamic factors (spatial distribution of source and sink habitats or underlying habitat suitability). In general, increasing heterogeneity in agricultural landscapes through introduction of refuges such as field margins, beetle banks or hedgerows is desirable. Even in areas highly exposed to insecticides, the non-target arthropods can be supported by ensuring enough source habitats, allowing populations to recover [26]. However, an important but often neglected factor here is the spatial distribution of insecticide application in a landscape. In farming systems with many scattered fields with intensively treated crops, the relative beetle populations are much more affected [21], even though local density in other areas can be high. This context needs to be taken into account when promoting farmland heterogeneity and crop diversity, and when determining both policy goals and the metrics used to measure policy impact. Once again, the specific context is important. Musters et al. [74] suggest that for flying insects the scale should be spatially large because of the dispersal ability, but for *Bembidion* the smaller landscape scale seems sufficient.

In conclusion, if we have a well-defined population target, the simulation approach used here provides a powerful tool to assess policy measures and their impact on key ecosystem service providers. These tools could be used to support innovative policy management towards goals such as those outlined in the Green Deal. However, care must be taken to properly define the specific metrics to be used to measure effects, and to understand the implications of changes in these metrics in population terms. Due to the degree of context specificity, once defined these metrics should not be taken out of the context of the key ecosystem service provider they are targeted towards. For example, *Bembidion* metrics should not be used to assess likely impacts of policy on pollinators.

## Supporting information

**S1 Appendix. Parametrization of ALMaSS landscape model for the Dutch landscapes.** (DOCX)

**S2 Appendix. Structural and farming heterogeneity of studied landscapes.** (DOCX)

**S3 Appendix. Collection of ALMaSS configuration files for simulation scenarios.** (ZIP)

**S4 Appendix. Results–supplementary data.**
(DOCX)

**S1 File. Results of beetle simulations for 10 Dutch study areas for DT$_{50}$ = 25 days (basic application rate of 1.22).**
(XLSX)

**S2 File. Results of beetle simulations for 10 Dutch study areas for DT$_{50}$ = 1.3 days (basic application rate of 41.79).**
(XLSX)

## Acknowledgments

This research was supported by the PBL Netherlands Environmental Assessment Agency through the project "Developing and application of a methodology to assess impacts of pesticides on key ecosystem services".

## Author Contributions

**Conceptualization:** Elżbieta Ziółkowska, Aaldrik Tiktak, Christopher J. Topping.

**Data curation:** Elżbieta Ziółkowska.

**Formal analysis:** Elżbieta Ziółkowska.

**Funding acquisition:** Elżbieta Ziółkowska.

**Investigation:** Elżbieta Ziółkowska.

**Methodology:** Elżbieta Ziółkowska, Christopher J. Topping.

**Project administration:** Elżbieta Ziółkowska.

**Software:** Christopher J. Topping.

**Supervision:** Aaldrik Tiktak, Christopher J. Topping.

**Visualization:** Elżbieta Ziółkowska.

**Writing – original draft:** Elżbieta Ziółkowska.

**Writing – review & editing:** Elżbieta Ziółkowska, Aaldrik Tiktak, Christopher J. Topping.

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
