## [Decision Letter · Decision Letter 0]

28 Aug 2022

PONE-D-22-13938Is the effectiveness of policy-driven mitigation measures on carabid populations driven by landscape and farmland heterogeneity? Applying modelling approach in the Dutch agroecosystemsPLOS ONE

Dear Dr. Ziółkowska,

Thank you for submitting your manuscript to PLOS ONE. After careful consideration, we feel that it has merit but does not fully meet PLOS ONE’s publication criteria as it currently stands. Therefore, we invite you to submit a revised version of the manuscript that addresses the points raised during the review process.

Your manuscript was examined by two reviewers both of whom indicated that the document was well written and relevant. They also indicated that with some revisions, it would be suitable for publication. However, they also expressed concerns. In both cases these were associated with the biology of the organisms and system. I encourage you to particularly consider these when you are revising your manuscript. I look forward your revised version.==============================

We look forward to receiving your revised manuscript.

Kind regards,

Sean Michael Prager, Ph.D.

Academic Editor

PLOS ONE

Journal Requirements:

"EZ work was supported by the PBL Netherlands Environmental Assessment Agency (https://www.pbl.nl/en) through the project “Developing and application of a methodology to assess impacts of pesticides on key ecosystem services” (contract no. 31134493)."

"EZ work was supported by the PBL Netherlands Environmental Assessment Agency (https://www.pbl.nl/en) through the project “Developing and application of a methodology to assess impacts of pesticides on key ecosystem services” (contract no. 31134493)."

6. We note that Figure 1 in your submission contain map image which may be copyrighted. All PLOS content is published under the Creative Commons Attribution License (CC BY 4.0), which means that the manuscript, images, and Supporting Information files will be freely available online, and any third party is permitted to access, download, copy, distribute, and use these materials in any way, even commercially, with proper attribution. For these reasons, we cannot publish previously copyrighted maps or satellite images created using proprietary data, such as Google software (Google Maps, Street View, and Earth). For more information, see our copyright guidelines: http://journals.plos.org/plosone/s/licenses-and-copyright.

Reviewers' comments:

Reviewer's Responses to Questions

**Comments to the Author**

1. Is the manuscript technically sound, and do the data support the conclusions?

Reviewer #1: Partly

Reviewer #2: Partly

2. Has the statistical analysis been performed appropriately and rigorously? 

Reviewer #1: I Don't Know

Reviewer #2: Yes

3. Have the authors made all data underlying the findings in their manuscript fully available?

Reviewer #1: No

Reviewer #2: Yes

4. Is the manuscript presented in an intelligible fashion and written in standard English?

Reviewer #1: Yes

Reviewer #2: Yes

5. Review Comments to the Author

Reviewer #1: Review for PONE-D-22-13938

This simulation study explored different factors influencing population viability of the carabid beetle Bembidion lampros in agricultural landscapes, specifically focusing on mitigation of adverse impacts of agricultural practices. Impacts of pesticide toxicity, reduction of spray drift, and increase in off-crop areas on beetle range and abundance were tested with a spatially explicit agent-based model, belonging to the ALMaSS family of models. The landscape component consisted of Dutch agricultural landscapes of differing complexity and uses, in the shape of 10 x 10 km maps. Their results suggest that options reducing direct beetle mortality, e.g., by reducing pesticide toxicity, in combination with increasing the number of margins would increase both the range and abundance of local populations. Authors suggest that simulation tools are essential for evaluating mitigation options and new environmental policy.

Overall, this is a nicely thought through and executed study and very relevant for current developments in environmental risk assessments and management. The main finding that reducing impacts on beetle survival and increasing field margins had the highest impact on population abundance and range is consistent with other studies exploring how perturbations of life history traits impacts population growth rate. In those studies, perturbations of survival rates consistently had the highest impact of population growth which is in line with the findings of this study, whereas here an additional factor was evaluated which essentially increased the capacity of the system. I agree with the authors about the role simulation tools should increasingly play in risk assessment and management and environmental policy and the authors provide a nice example of how this could be done. The manuscript is written well, but what I am missing is sufficient information on the biology and ecology of the modeled species which prevents from fully evaluating the analysis and results. The reader should be at least somewhat familiar with the basic ecology and habitat use of the modeled species, yet not even rudimentary descriptions have been provided. Rather the authors refer to old publications and to the model documentation which I checked, but it still does not provide enough relevant information on the species. Please find more specific comments below.

LN 22 - Bembidion

LN 116-126 - I can appreciate the decades of work that went into developing ALMaSS and given its complexity there is no really practical way of fully describing it in each and every publication. The model documentation should be a repository of all relevant information on the conceptual model, as well as on model testing and validation. I found the amount of information in the ODdox underwhelming and the reference to the Bilde&Topping 2004 paper not sufficient. That paper does not contain a detailed account of species biology and how it was implemented, the model evaluation is not included either. I would strongly urge the authors to provide a bit more information on the species, either in section 2.2 or in the Appendix, specifically on its life cycle, life history traits, habitat use, population regulation mechanisms, and dietary preferences. And of course, how such information was implemented in the model. This type of document needs to be produced once and can be re-used for any new publication.

LN 223 - 10 x 10 km, not km2

LN 245 - Please provide a more information, ie the full function, for intraspecific predation.

LN 247 – 255 - I understand this is a case study and not meant as a realistic assessment of risk from pesticides, but the implementation of effects is very simplistic and at odds with the much more elaborate implementation of exposure in the landscape.

LN 283 - Always better to pull the actual parameter values out of equations and describe/define them in text.

LN 336-365 - In order to make this study somewhat reproducible, you should provide initial state and conditions of the model, as well as the scripts used for this specific study. You should also provide all the simulation data that are presented in this manuscript and ideally the scripts used for analysis and plotting. It doesn’t look like any of this was provided, making this study completely irreproducible.

LN 373-375 - Did you conduct any testing to support the claim about very small effect when a longer half-life was applied?

LN 401-405 - It is really hard to understand the impacts of different farm types when we weren’t provided enough information on habitat use by the modeled species.

LN 429 - Information about baseline dynamics and variation across replicates should be more explicit and provided at the beginning of the results section.

Reviewer #2: General comments

I have had the opportunity to review this manuscript, model the impacts of mitigation measures on carabid populations with the hope of influencing policy decisions. The authors report interesting results in that pesticide toxicity seems to be the most influential factor impacting carabids, although interactions with other landscape variables were noted. I note that the manuscript is fairly well-written, has good flow, and requires little editing in that regard. Most of my revisions address transparency in the model and its assumptions, and several questions regarding further details on the study. Given the limited scope of the study, with most of the model conditions applying only to Dutch farming methods and policy, the manuscript may be a better fit for a more localized journal such as the European Journal of Entomology.

My main criticism of the study is that, as with most modeling ventures, the model is only as good as the assumptions made in it. In this particular case, little attention was given to the biology of the “model” species the authors chose. Little to no data or background is provided about the biology and ecology of this species. While I do agree it is abundant in the Netherlands, it must be pointed out it is classified primarily as a heathland species, and not one of major agricultural habitation (see Turin et al., 2022 – The Ecology and Conservation of Dutch Ground Beetles). The authors really provide no detail on why it was chosen as a model species, other than it is abundant and had been used in this type of model previously. However, because of the importance of ecosystem services to policy decisions, simulations should have included granivorous or omnivorous species as well, such as genera of Amara or Harpalus. Using a model species that is carnivorous missed important potential ecosystem services and would suggest the species is, perhaps, not truly a good model species for this modelling scenario. Several assumptions in the model are also problematic, as suggested below.

A second criticism is that the model was not validated, nor was a sensitivity analysis done on model parameters. Without such procedures, it is impossible to know how sensitive model parameters are to changing input. Likewise, a validation would provide confidence in the predictions of the model. If the authors have these data, they are encouraged to present them

My final comment concerns the discussion section, which nicely describes and contextualizes the results. However, important caveats of the study must be presented, including the fact that only a single, carnivorous carabid was used, model parameters were not tested for sensitivity, and no validation was done on model output. Another critical caveat of the model is that it treats all crops equally when it comes to pesticide exposure but in reality, crop canopies differ markedly among crops. As such, pesticide penetration to the lower canopy and thus, the habitat of the ground beetle, will be different among crops. This detail is important, but not considered in the study.

Specific comments:

At first mention of B. Lampros, mention that it is carnivorous. Clarify why omnivorous/granivorous carabid species were not included as the model species. Of particular importance as well is the fact B. Lampros is classified as a heathland species, and not one of particular importance in cropped fields (Turin et al. 2022). This contrasts with the objects of the model and makes it an odd choice for a model species.

L99 – provide nomenclature of species at first mention – in this case for B. Lampros, it is Herbst, 1784)

L133 – I believe manuscript would benefit from a schematic showing the model and its parameters, as well as the process and statistics used in the model

L240 – The assumption that males do not limit population size is erroneous. Populations cannot grow without males present, so how can a female only population be a realistic scenario upon which to build a model for policy decisions?

L270 – Please provide a citation to show there is no impact of fungicides on carabid beetles

L290 – beetles were exposed to a pesticide for a period defined by at least 1-25 days. However, carabids are highly mobile and therefore would not be exposed to the same level of toxicity over time. They could move out of the field, move to an area where the canopy intercepted more of the pesticide, etc. I don’t understand how they can be modeled to have continuous exposure for 25 days.

L327 – the authors do not define what is a replicate in this study

L365 – Which R package was used here and how were the analyses conducted? Please provide more depth

L447 – I don’t tend to agree with this. As you state in the previous section, the results were mainly due to the large impact of pesticide toxicity and not the impact of field margin. It is likely that the effects of pesticide toxicity masked those of field margin, but this possibility seems not considered here.

L461 – what is meant by “realistically simulate”? The model is based on assumptions that are unlikely to be true for other carabid species, and has provided limited justification for including B. lampros as the model species in this study. Clarification is needed here

L468 – Are these measures actually relevant at the EU level? Please provide evidence for this. Member countries currently have and continue to implement their own guidelines with full autonomy. Again, the results obtained here are based on a simulation with Dutch conditions.

L485 – this paragraph nicely addresses the caveat for animal farms in mixed farming. It is prudent to also include a caveat that crop architecture was considered equivalent for all crops in the model, which may provide poor estimates of pesticide sensitivity to due interception by foliage. In addition, provide a caveat that beetles were assumed to be largely stationary and thus, the model makes not provision for immigration or emigration of beetles into a particular field or cell.

L583 – this is not true. B. lampros is wing dimorphic and disperses by flight (Turin et al, 2022) and thus, the smaller landscape scale is not sufficient and at worst, potentially erroneous.

L585 – I don’t disagree with the conclusion, but it must be pointed out that the assumptions and biology of the model are not necessarily realistic. I realize this may preclude publication, but this can be handled by re-running the model based on more realistic biological parameters.

L592 – most certainly Bembidion metrics should not be used to assess impacts on pollinators, the question is should the even be used to assess impacts on other carabids? This should be justified here.

Figure 4 – please show the SE on each bar, and include the names of the 10 study areas in the figure caption.

Figures 5-7 – It seems this data would be better represented in a series of heatmaps, where 1 cell of the heatmap would be used to show the results in each study area.

Several editorial comments appear in the attached document.

6. PLOS authors have the option to publish the peer review history of their article (what does this mean?). If published, this will include your full peer review and any attached files.

Reviewer #1: No

Reviewer #2: No

---

## [Author Response · Author response to Decision Letter 0]

12 Oct 2022

We appreciate the careful review and constructive suggestions. We provide our detailed response to the reviewers’ comments in a separate document.

---

## [Decision Letter · Decision Letter 1]

4 Nov 2022

PONE-D-22-13938R1Is the effectiveness of policy-driven mitigation measures on carabid populations driven by landscape and farmland heterogeneity? Applying a modelling approach in the Dutch agroecosystemsPLOS ONE

Dear Dr. Ziółkowska,

Thank you for submitting your manuscript to PLOS ONE. After careful consideration, we feel that it has merit but does not fully meet PLOS ONE’s publication criteria as it currently stands. Therefore, we invite you to submit a revised version of the manuscript that addresses the points raised during the review process.

Thank you for your revised submission. Once again this has been reviewed by two qualified individuals. While they both indicated that some revisions were substantial and appreciated, they also still voiced concerns. In particular, both indicate concerns about the lack of sensitivity analyses. One also voiced some concerns about the level of support relative to some claims. I encourage you to consider both these topics closely and address them in a resubmission.

We look forward to receiving your revised manuscript.

Kind regards,

Sean Michael Prager, Ph.D.

Academic Editor

PLOS ONE

Reviewers' comments:

Reviewer's Responses to Questions

**Comments to the Author**

1. If the authors have adequately addressed your comments raised in a previous round of review and you feel that this manuscript is now acceptable for publication, you may indicate that here to bypass the “Comments to the Author” section, enter your conflict of interest statement in the “Confidential to Editor” section, and submit your "Accept" recommendation.

Reviewer #1: (No Response)

Reviewer #2: (No Response)

2. Is the manuscript technically sound, and do the data support the conclusions?

Reviewer #1: Yes

Reviewer #2: No

3. Has the statistical analysis been performed appropriately and rigorously? 

Reviewer #1: N/A

Reviewer #2: Yes

4. Have the authors made all data underlying the findings in their manuscript fully available?

Reviewer #1: Yes

Reviewer #2: Yes

5. Is the manuscript presented in an intelligible fashion and written in standard English?

Reviewer #1: Yes

Reviewer #2: Yes

6. Review Comments to the Author

Reviewer #1: Thank you for addressing my comments.

I see from other comments that the question of sensitivity analysis was brought up - and even though I can see the point of the authors that this model has been thoroughly analyzed elsewhere, I think it would be good to hear from the authors what they think how the assumptions about the biology, especially movement, influence the simulation results. There should be a section in the discussion that addresses this, perhaps including the main findings from previous analyses, as we don't expect readers to dive in all the previous publications, but would still like to be informed about the analyses relevant for the simulations presented.

Reviewer #2: I thank the authors for their many edits, which have improved the clarity of the manuscript in many ways. I still have several questions with regard to the manuscript, however. Many of these remain poorly addressed in my opinion.

While the authors provide more evidence for the use of their chosen species, their conclusions are wide-ranging and do not reflect the fact only a single beetle species was used. Should policy be based on a model tested with only a single species? Moreover, L30-31 suggest results were highly context-specific, which they are likely to be carabid species as well. Yes, they have used a species that is in the top 25 most trapped in ag fields, but species #25, if it was, is a substantial difference from the top 10 species. L32-34 suggests this model as a powerful too to support policy, but how can we conclude this is so when only a single carabid species was used? Until the model can be proved to be non taxon-specific, (it is currently contextually-specific), I remain on the sidelines as to the validity of its predictions and use in policy development. I believe the authors can accommodate this is the markedly tone down the conclusions and add the caveats suggested by the reviewers.

In addition, the model needs a sensitivity analysis done when it is reparameterized, was done here. I understand this was done upon completion of the model in previous work, but readers cannot have confidence in the model unless it is shown just how sensitive the parameters are to the starting values of model runs. This is especially true for the canopy effects (pesticide interception), where data were not tested and were not included from field scenarios. If this parameter is overly sensitive, it needs data and assumptions are not enough to produce accurate model results. I would more happily accept their argument on this parameter if I could be assured it was not that sensitive and as such, their absence of data on the subject would have little impact on the overall model predictions.

I appreciate the citation of the Bilde and Topping (2004) publication but I think this manuscript would benefit from even brief description of the model schematic and its parameters. Readers should not have to go to another publication to understand the model being used here, and this can be easily added in a supplemental file.

The author response to my initial comments on point #15, regarding realistic simulations, is mostly valid. I do agree that their simulations apply to all beetles with similar life history, but the agro ecosystem is rife with species that have a similar life history to the species used here but that may feed at different trophic levels and as such, the impacts of the model may be different for the agroecosystem as a whole based on model predictions. For example, would omnivorous or granivorous respond similar to model predictions, where these species have similar life histories? Again, it may be useful to couch the conclusions in this regard, and limit them to the species studied and point out conclusions may vary depending on species used.

7. PLOS authors have the option to publish the peer review history of their article (what does this mean?). If published, this will include your full peer review and any attached files.

Reviewer #1: No

Reviewer #2: No

---

## [Author Response · Author response to Decision Letter 1]

6 Dec 2022

The response to specific reviewers' comments is provided in a separate document.

---

## [Decision Letter · Decision Letter 2]

12 Dec 2022

Is the effectiveness of policy-driven mitigation measures on carabid populations driven by landscape and farmland heterogeneity? Applying a modelling approach in the Dutch agroecosystems

PONE-D-22-13938R2

Dear Dr. Ziółkowska,

We’re pleased to inform you that your manuscript has been judged scientifically suitable for publication and will be formally accepted for publication once it meets all outstanding technical requirements.

Kind regards,

Sean Michael Prager, Ph.D.

Academic Editor

PLOS ONE

Additional Editor Comments (optional):

Reviewers' comments:

Reviewer's Responses to Questions

**Comments to the Author**

1. If the authors have adequately addressed your comments raised in a previous round of review and you feel that this manuscript is now acceptable for publication, you may indicate that here to bypass the “Comments to the Author” section, enter your conflict of interest statement in the “Confidential to Editor” section, and submit your "Accept" recommendation.

Reviewer #1: All comments have been addressed

2. Is the manuscript technically sound, and do the data support the conclusions?

Reviewer #1: Yes

3. Has the statistical analysis been performed appropriately and rigorously? 

Reviewer #1: N/A

4. Have the authors made all data underlying the findings in their manuscript fully available?

Reviewer #1: Yes

5. Is the manuscript presented in an intelligible fashion and written in standard English?

Reviewer #1: Yes

6. Review Comments to the Author

Reviewer #1: Dear authors,

Thank you for addressing my comments. This is now a much clearer and more informative manuscript.

7. PLOS authors have the option to publish the peer review history of their article (what does this mean?). If published, this will include your full peer review and any attached files.

Reviewer #1: No

---

## [Editor Report · Acceptance letter]

14 Dec 2022

PONE-D-22-13938R2 

Is the effectiveness of policy-driven mitigation measures on carabid populations driven by landscape and farmland heterogeneity? Applying a modelling approach in the Dutch agroecosystems 

Dear Dr. Ziółkowska:

I'm pleased to inform you that your manuscript has been deemed suitable for publication in PLOS ONE. Congratulations! Your manuscript is now with our production department. 

Kind regards, 

on behalf of

Dr. Sean Michael Prager 

Academic Editor

PLOS ONE